# Intolerance of Uncertainty and Mindfulness as Determinants of Anxiety and Depression in Female Students

**DOI:** 10.3390/bs9120135

**Published:** 2019-12-03

**Authors:** Marina Nekić, Severina Mamić

**Affiliations:** 1Department of Psychology, University of Zadar, Obala kralja Petra Krešimira IV. 2, 23000 Zadar, Croatia; 2Student Counselling Centre, University of Zadar, Trg kneza Višeslava 9, 23000 Zadar, Croatia

**Keywords:** intolerance of uncertainty, mindfulness, anxiety, depression, female students

## Abstract

Bearing in mind the characteristics of an academic setting, as well as the developmental tasks young people inevitably face, there is a need to identify and study the factors that play an important role in the development and maintenance of psychological distress among college students. One factor that has emerged as crucial for the development of both anxiety and depression is the intolerance of uncertainty. On the other hand, there appears to be mounting evidence that mindfulness is an important factor that can be beneficial for the psychological health of college students. Taking this into consideration, the purpose of the current study was to determine the contribution of the intolerance of uncertainty and mindfulness in explaining the symptoms of depression and anxiety in a sample of female college students (*n* = 282) whose average age was 21. The results indicated that students had moderate levels of anxiety and depression. Additionally, they were, on average, intolerant of uncertainty and mindfulness. Two hierarchical regression analyses revealed that the intolerance of uncertainty and mindfulness significantly predict both anxiety and depression in female students, and that mindfulness partially mediates the relationship between the intolerance of uncertainty and anxiety/depression. The results are discussed in the light of previous studies, and its clinical implications.

## 1. Introduction

Academic settings have been recognized as very demanding and challenging for the individual. Apart from emphasizing success and performance, going to college represents an important shift from high school, where everything was much more structured and stable. College students are expected to be more independent and self-sufficient, yet they usually face financial difficulties, separation from their social network, and the challenges of creating new relationships [1]. Developmentally speaking, the period from 18 to 29 years (i.e., adolescence and young adulthood) is known as emerging adulthood. It is a period of many psychological and psychosocial changes which, combined with previously the aforementioned college demands, also means much pressure and changes in one’s life [2]. If students adapt poorly to this new setting in terms of setting unreal expectations, and becoming preoccupied with the mistakes they make, they are likely to develop certain mental health issues [1]. It is important to mention that many psychological disorders onset by the age of 24, which overlaps with the years spent at college [3]. Blanco et al. [4] report that almost half of college students meet DSM-IV criteria for at least one mental disorder in the previous year, with almost twelve percent of them suffering from anxiety, and seven percent from depression. They also found that less than twenty percent of those suffering from anxiety have received treatment, and about thirty-five of those who suffer from mood disorders. Furthermore, ninety-five percent of counselling center directors reported a significant increase in the number of students with mental health problems [5]. According to a recent study, almost one third of college students have symptoms of anxiety and depression [6]. The initial results from The WHO World Mental Health Surveys International College Student Project (WHO WMH-ICS) have shown that thirty-five percent of almost 14,000 college students from 19 colleges and 8 countries report at least one mental disorder. Furthermore, one of the socio-demographic variables that emerged as being one of the positive correlates of the lifetime and 12-month prevalence of mental disorders is female gender [7]. Taking all into consideration, it is of crucial importance to identify the risk factors for the development of psychological disorders among female students. Their identification can lead to the development of new techniques and treatments for these disorders. One factor that emerges as being important for the development of both anxiety and depression is the intolerance of uncertainty (IU). 

### 1.1. IU and Psychological Well-Being

IU is broadly defined as a tendency to be fearful of everyday uncertain situations [8], and to interpret every ambiguity as something stressful and frustrating [9]. More recently, Carleton [10] (p. 31) defined it as “an individual’s dispositional incapacity to endure the aversive response triggered by the perceived absence of salient, key, or sufficient information, and sustained by the associated perception of uncertainty.” The main reason why this kind of thinking and behaving is not adaptable is because it interferes with problem-solving i.e., a person reacts impulsively to decrease uncertainty which can be efficient for some time, but does not lead to problem solving. Also, this pattern of thinking and behaving generates dysfunctional emotional states, and inhibits problem-focused behavior [11]. In order to better understand this construct, studies have focused on its factor structure, and shown that IU is best explained by two dimensions: Prospective and inhibitory anxiety. Prospective anxiety (PA) represents the cognitive dimension, i.e., it refers to the desire for predictability, for knowing what will happen in the future, and engagement in seeking information to reduce uncertainty. On the other hand, inhibitory anxiety (IA) represents the behavioral dimension; avoidance and paralysis when faced with uncertainty [12]. A certain amount of uncertainty is expected and normal, and is present in every stage and domain of life. One can feel uncertain about their career, romantic partners, and future goals, but still feel in control. However, to some people, this can be extremely overwhelming, and become a source of psychological distress [13]. IU has a central role in Dugas, Gagnon, Ladouceur, and Freestone´s [14] cognitive-behavioral model of GAD, which has strong empirical support. However, studies have shown that IU is a transdiagnostic vulnerability factor for the development and maintenance of many psychological disorders [9,10,15]. Because academic settings can often be unstable and uncertain and can be overwhelming and extremely stressful to some [1], it is clear that IU can be one of the risk factors for the development of mental health problems in this population.

### 1.2. Mindfulness as a Protective Factor

One construct that emerged as one of the protective factors for the development of many psychological disorders, and is extensively examined in recent studies, is mindfulness [16,17]. Commonly defined as the state of attention and awareness in the present moment, it found its way into the western world, although it is rooted in the teachings of Buddha [18]. Being mindful may seem simple to achieve, but the state of mindfulness contradicts our everyday functioning, which can be described as non-attentive [19]. Moreover, when given the opportunity to spend 6 to 15 min by themselves and do nothing, college students reported that they did not enjoy the experience, but did enjoy doing mundane activities, even if that meant administering electric shocks to themselves [20].

According to Shapiro et al. [21], intention, attention, and attitude (IAA) are three fundamental components of mindfulness which occur simultaneously and lead to a shift in perspective they call *reperceiving.* Reperceiving is a meta-mechanism that helps a person disengage from everyday dramas and simply stand back and witness them [21]. Mindfulness has been shown to have a beneficial effect on numerous negative states and psychological health problems [16,22], and is negatively associated with IU [23]. When it comes to academic settings, mindfulness has been shown to have a positive impact on both the academic and personal level [24,25].

### 1.3. The Present Study

Considering the aforementioned studies, as well as a lack of research into both IU and mindfulness in Croatia, the objective of this research is to determine the contribution of IU and mindfulness in explaining the symptoms of depression and anxiety among female college students. 

## 2. Materials and Methods

### 2.1. Participants

The research was conducted on 282 female students from different Universities in Croatia. Most of the sample consisted of students of social (*N* = 141, 50%), technical/natural sciences (*N* = 46, 16%), and humanities (*N* = 95, 34%), and of the average age of 21 (*SD* = 2.01; age range: 18–32). Most of them were undergraduates (*N* = 185, 66%), and the rest of the sample was at the graduate level (*N* = 97, 34%). The participants estimated that they were neither satisfied nor dissatisfied with their health as well as their academic achievements. They also evaluated the previous year as stressful. These data were collected solely for the purpose of the description of the participants and were not used in further analyses. All participants gave their informed consent for inclusion before participating in the study. The Ethical Committee of the Department of Psychology of University of Zadar approved the research protocol, and furthermore, the study was conducted in accordance with the Declaration of Helsinki.

### 2.2. Measurement Variables and Instruments

*Demographic questionnaire.* Female students were asked to provide general information about their age, level of study, and major. In this part of the questionnaire, the participants gave their evaluation of satisfaction with general health conditions, academic achievement, and the stressfulness of the previous year. All aforementioned variables were assessed by one item and the participants responded on a five-degree scale (from 1—totally dissatisfied to 5—totally satisfied). The estimation of the stressfulness of the previous year also included a five-degree scale, where 1 was not stressful, while 5 was extremely stressful.

*Intolerance of uncertainty.* Intolerance of Uncertainty Scale—IUS-11 is a Croatian adaptation [26] of the Serbian version [27] of Freeston et al. [11] original scale of 27 items. This adapted version has 11 items which are used to assess a person’s tendency to react negatively to uncertain and ambiguous situations. The task of the participant is to rate the degree to which each of the items (e.g., "Unforeseen events upset me greatly”) applies to them on a 5-degree scale (from 1 = not at all characteristic of me to 5 = entirely characteristic of me). In this research, the factor structure of the scale confirmed two factors, PA (“One should always look ahead so as to avoid surprises”) and IA (“The smallest doubt can stop me from acting”) (*χ*2 = 116.01, *p* < 0.01, *df* = 42, *χ*2/*df* = 2.76, CFI = 0.96, TL I = 0.95, RMSEA = 0.07, SRMR = 0.04). The IUS-11 has demonstrated excellent internal consistency (α = 0.91). Cronbach’s internal reliability coefficient for the PA was 0.89, and for IA, 0.86. A higher score on each of the subscale indicates a higher degree of PA or IA. The sum of the results from both subscales can be used as an indicator of IU, so a higher result means more intolerance. 

*Mindfulness*. Mindful Attention Awareness Scale–MAAS [18,28] is a measure that consists of 15 statements that examine the experiences of an individual in everyday life, as well as variations in the awareness of action, interpersonal communication, thoughts, emotions, and various physical states. Participants are asked to indicate on a six-degree scale from 1 (almost always) to 6 (almost never) to what extent each statement (e.g., "I find it difficult to stay focused on what’s happening in the present") refers to their real experiences, not those that they think they should have. A higher score on this scale reflects higher levels of mindfulness. Researches regularly point to a one-factor structure of the scale, originally provided by Brown and Ryan [18], and the same was obtained in this research (*χ*2 = 279.83, *p* < 0.01, *df* = 89, *χ*2/*df* = 3.14, CFI = 0.87, TLI = 0.85, RMSEA = 0.08, SRMR = 0.06). In addition, a high coefficient of internal consistency was also confirmed in current research (α = 0.85).

*Anxiety and Depression*. Depression, Anxiety and Stress Scale (DASS-21) [29,30] is used to assess anxiety and depression levels in female students. This is one of the most known measures for examining levels of anxiety, depression, and stress. The scale consists of a total of 21 statements, i.e., each subscale consists of seven statements, and a higher score on each of them indicates higher levels of anxiety, depression, and stress. For the purpose of this research, only anxiety (e.g., “I was worried about situations in which I might panic and make a fool of myself”) and depression (e.g., “I felt that I had nothing to look forward to”) subscales were used (*χ*2 = 40.66, *p* < 0.01, *df* = 13, *χ*2/*df* = 3.13, CFI = 0.96, TLI = 0.94, RMSEA = 0.08, SRMR = 0.04). Participants were asked to indicate on a 4-degree scale (from 0 = did not apply to me at all to 3 = applied to me very much or most of the time) how often they have experienced the condition described in the statement. Cronbach’s internal reliability coefficient for the anxiety subscale was 0.84, and for depression was 0.89. Authors of this scale recommend the cut-off scores categorizing the severity of depression, anxiety, and stress into five levels (normal, mild, moderate, severe, and extremely severe) [29]. 

### 2.3. Procedure

The research was conducted over the online link that was posted on the official social network pages of the university students’ groups. At the beginning of the questionnaire, there were instructions as well as the general description of the aim of the research and informed consent. It was emphasized that the results would be analyzed at the group level. The average time for completing the questionnaire was about 15 min. The questionnaire was completed anonymously, and no compensation was given to students. 

### 2.4. Analysis Plan

Firstly, descriptive statistics of measured variables were presented. Prior to any statistical analyses, we tested whether the results deviate from the normal distribution in order to use parametric data analysis. The results showed the Kolmogorov–Smirnov tests were significant, however, the skewness indexes (−0.54–0.85) and the kurtosis indexes (−0.50–0.46) for all measures were acceptable (skewness < 3, kurtosis < 8) [31]. Afterwards, the matrix of correlations was presented, and the results of two hierarchical regression analyses with anxiety and depression as a criterion variable, and with PA or IA as dimensions of IU and mindfulness as predictor variables. We also used Baron and Kenny [32] four steps for establishing mediation with the assumption that mindfulness was a mediator and dimensions of IU were predictors. Data were analyzed using 13.3 version of Statistica, and 6.12 version of M Plus. 

## 3. Results

### 3.1. Descriptive Parameters for IU (PA and IA), Mindfulness, Anxiety and Depression in Female Students

As shown in Table 1, the results of IU (whether it is a global result obtained on that scale or the result on its two subscales) are slightly shifted towards lower values (*M* = 2.70, *SD* = 0.82), which means that female students have below average tolerance for uncertain and ambiguous situations.

Female students in this study perceived their ability to purposely focus attention to here and now nonjudgmentally slightly above average (*M* = 3.87, *SD* = 0.75). Since the sample of our study is a non-clinical, the results of anxiety (*M* = 6.21, *SD* = 4.47) and depression (*M* = 6.44, *SD* = 5.16) were expected to be shifted more on lower values. Based on cut-off scores suggested by Lovibond and Lovibond [29], results in our study showed that levels of anxiety and depression are between mild and moderate. 

### 3.2. Correlation Between Anxiety and Depression with IU and Mindfulness among Female Students

In Table 2, correlation coefficients between measured variables are presented. As shown, all coefficients of correlation were statistically significant on *p* < 0.001. The average value of correlations was moderate, and ranged from 0.38 to 0.65. There was a statistically significant positive association of anxiety with IU (*r* = 0.43, *p* < 0.001), IA (*r* = 0.38, *p* < 0.001) and PA (*r* = 0.41, *p* < 0.001), and depression (*r* = 0.58, *p* < 0.001), as well as a significant negative association with mindfulness (*r* = −0.43, *p* < 0.001). Also, depression followed the same pattern of correlation with IU (*r* = 0.48, *p* < 0.001) (IA and PA) and mindfulness (*r* = −0.44, *p* < 0.001) as anxiety. Furthermore, IU was statistically negatively associated with mindfulness (*r* = −0.42, *p* < 0.001). In addition, it can be said that female students who are more anxious and depressed are less mindful and have less tendencies to tolerate uncertainty. 

### 3.3. IU and Mindfulness as Predictors of Anxiety and Depression

Table 3 shows the results of HRA with anxiety as the criterion, PA and IA as the predictors, and mindfulness as the mediator. In the first step, IU dimensions were entered. Nineteen percent variance of anxiety (F = 33.70, *p* < 0.001) was explained with these two predictors, with IA as the strongest predictor (β = 0.30, *p* < 0.001). In the second step, mindfulness (β = −0.30, *p* < 0.001) was entered so afterwards the contribution of PA was reduced, and became insignificant (β = 0.12, *p* < 0.06) while IA remained significant, but value of beta ponder decreased (β = 0.21, *p* < 0.001). A total of 27% of anxiety in female students was explained with IU and mindfulness (F = 34.18, *p* < 0.001). Mindfulness proved to be a significant partial mediator of the relationship between dimensions of IU and anxiety, and also contributed to the explanation with an additional 8% of anxiety in female students. 

In Table 4, the results of HRA with depression as the criterion were presented. As in the previous HRA, we used the same predictors and mediator. Therefore, in the first step, IU dimensions were entered. The 24% variance of depression (F = 43.43, *p* < 0.01) was explained with these two predictors, with IA as the stronger predictor (β = 0.33, *p* < 0.01) than PA (β = 0.20, *p* < 0.01). In the second step, mindfulness (β = −0.29, *p* < 0.01), was entered so that the contribution of dimensions of IU were reduced later (PA, β = 0.14, *p* < 0.03; IA, β = 0.25, *p* < 0.00). A total of 31% of depression (F = 41.39, *p* < 0.00) was explained with IU and mindfulness. In this case, mindfulness once again proved to be a significant partial mediator of the relationship between dimensions of IU and depression, with the contribution of an additional 7% to regression coefficient. 

## 4. Discussion

In recent years, there has been an increasing interest in identifying mental disorders among college students. According to data from one extensive study by WHO, 1/3 of first-year students had at least one mental disorder, with female gender as a meaningful correlate of both the lifetime and 12 month prevalence of mental disorders [7]. Because of the severity of these findings, as well as an apparent lack of research into the certain factors that could play a significant role in the development of mental disorders among students, in this study we sought to examine the contribution of IU and mindfulness in explaining the symptoms of depression and anxiety in a sample of female college students. 

When it comes to the relationship between anxiety and IU, whether we are speaking of the result on a global scale or the results on two subscales, there was a positive association between these constructs. This is in line with numerous studies conducted on various samples [11,14]. However, the relationship between depression and IU has not received much attention thus far. According to Miranda, Fontes, and Marroquín [33], the reason why IU may play an important role in the development and maintenance of symptoms of depression is the fact that IU works as a cognitive bias in terms that the person develops a global negative prediction towards future events. One of the explanations could be the fact that high levels of rumination (which is a central component of depression) magnify the negative effects of uncertain everyday situations, which in turn leads to a stronger belief that negative events will happen when faced with ambiguity [13].

As expected, mindfulness was negatively associated with anxiety and depression as well as IU and its two components. In other words, those who feel like *running on automatic* become aware of emotions and physical tension only when they became obvious, and have difficulties with focusing on events happening in the present moment, are also more likely to have symptoms of anxiety and depression. This corroborates previous findings where higher levels of mindfulness, whether defined as a trait or skill that can be cultivated, mean fewer mental health problems [22]. Furthermore, Alimehdi et al. [16] found that mindfulness-based stress reduction programs decreased the symptoms of GAP, but also IU and anxiety sensitivity, its two key components. The relevance of mindfulness in overall well-being is clearly supported by prior studies, but also with current research. To be more specific, IU and mindfulness emerged as reliable predictors of both anxiety and depression among female students since they explained 27% of the variance of anxiety, and 31% of depression. Furthermore, mindfulness partially mediated the relation between the dimensions of IU and anxiety, as well as the relation between the dimensions of IU and depression. In other words, the inability of individuals to remain calm when faced with uncertainty can predict the onset of anxiety/depression, and this process is partially due to their inability to decenter from negative thoughts and sensations. The reason why mindfulness could be beneficial for mental health could be *reperceiving*, which was explained earlier in the paper [21]. According to *Mindfulness-to-Meaning Theory*, increases in mindfulness during meditation were associated with the more frequent use of positive reappraisal (a positive psychological process during which some stressful events are reconstructed as more positive and meaningful). These two processes mutually enhance each other, and are part of a larger upward spiral [34]. It is also possible that mindfulness buffers the influence of Behavioral Inhibition System sensitivity on the development of mental health problems [17]. The correlation the nature of research has well known limitations, as well as a relatively small sample size. Surely, it would be useful to test the assumed causal relations between these constructs. In this regard, we consider that the study of Johnstone et al. [35] could be a good starting point for future experimental studies. In a recent study, Auerbach et al. [7] found that female gender is a risk factor for mental health issues. Overall, Soysa and Wilcomb [36] emphasized that given the gender differences in psychological outcomes, female students are more vulnerable. However, male students have to be considered in this type of research since UK National Statistics [37] has shown that male students are at greater risk of suicide. Therefore, future research should include the protective and risk factors of the mental health of male and female students, as well as across different age groups. Taking into account Galante’s et al. [25] findings, mindfulness training for students could be an effective foundation for the mental health of college students. In addition, there are significant and practical implications of this research, primarily because of the emphasized need to incorporate mindfulness training in student counselling centers, as Leland [38] recommended in his article.

## Figures and Tables

**Table 1 behavsci-09-00135-t001:** Descriptive statics for measured variables.

Measured Variables	M	SD	Range
Intolerance of uncertainty (IU)	2.70	0.82	1–5
Prospective anxiety (PA)	2.65	0.94	1–5
Inhibitory anxiety (IA)	2.74	0.86	1–5
Mindfulness	3.87	0.75	1.5–5.9
Anxiety	6.21	4.47	0–21
Depression	6.44	5.16	0–21

**Table 2 behavsci-09-00135-t002:** Matrix of correlation.

Variables	IU	PA	IA	M	A
Intolerance of uncertainty (IU)					
Prospective anxiety (PA)	0.90				
Inhibitory anxiety (IA)	0.91	0.65			
Mindfulness	−0.42	−0.39	−0.37		
Anxiety	0.43	0.41	0.38	−0.43	
Depression	0.48	0.46	0.41	−0.44	0.58

Note: All coefficients of correlation are significant on *p* < 0.001.

**Table 3 behavsci-09-00135-t003:** Results of hierarchical regression analysis with IU and mindfulness as predictors and anxiety as criterion.

Variable	β	SE β	R^2^	R^2^ Change
Step 1			0.19 **	
Prospective anxiety (PA)	0.19	0.07 *		
Inhibitory anxiety (IA)	0.30	0.07 **		
Step 2			0.27 **	0.08 **
Prospective anxiety (PA)	0.12	0.06		
Inhibitory anxiety (IA)	0.21	0.07 **		
Mindfulness	−0.30	0.06 **		

* *p* < 0.01; ** *p* < 0.001.

**Table 4 behavsci-09-00135-t004:** Results of hierarchical regression analysis with IU and mindfulness as predictors and depression as criterion.

Variable	Β	SE β	R^2^	R^2^ Change
Step 1			0.24 **	
Prospective anxiety (PA)	0.20 **	0.07		
Inhibitory anxiety (IA)	0.33 **	0.07		
Step 2			0.31 **	0.07 **
Prospective anxiety (PA)	0.14 *	0.07		
Inhibitory anxiety (IA)	0.25 **	0.07		
Mindfulness	−0.29 **	0.05		

* *p* < 0.01; ** *p* < 0.001.

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
