# Peer review of "Intolerance of Uncertainty and Mindfulness as Determinants of Anxiety and Depression in Female Students"

_behavsci, 2019, doi:10.3390/bs9120135_

Round 1

Reviewer 1 Report

This is a noteworthy paper, which aim was to determine the contribution of intolerance of uncertainty and mindfulness in explaining the symptoms of depression and anxiety in a sample of female college students (n = 282) aged 21.
Title consistent with the problem actually presented and reflect the main message of the study.
Article possess scientific and practical value, it is very interesting and well conducted.
Abstract give an adequate picture of the entire article. The aim of the paper is clear and was achieved. The interpretation of the results is clearly presented. The sample size is adequate and representative.
Applications are not very clearly written. Please correct.
Could the authors write more detail the practical aspect of research?

Reviewer 2 Report

The present manuscript is focused on a serious health problem, and provides interesting data about some risk factors related to anxiety and depression in female students. The results obtained highlight the role of variables intolerance of uncertainty and mindfulness, and could be useful for developing more effective prevention programs. The manuscript is properly elaborated, and only some minor questions should be improved by authors:

Abstract

The authors should indicate “average age 21”, since not all participants were aged 21.

Introduction

Introduction is well organized and the different subsections are adequate. The aim of the research is clearly established by the authors.

However, the review of previous research should be expanded by authors. There are numerous previous studies on anxiety and depression in college students that could be included in the introduction to expand the contextualization of this study.

In the subsection “1.1. IU and Psychosocial Well-being”, it is indicated that Intolerance of Uncertainty is not adaptable because this kind of thinking and behaving interferes with problem-solving, “i.e. person reacts impulsively to decrease uncertainty which can be efficient for some time, but does not lead to problem solving [9].” However, in subsection 2.2. Measurement Variables and Instruments (lines 108-119), the scale of Intolerance of Uncertainty include one factor related to inhibition of action (IA). I think that authors should explain better these different negative behaviors related to IU, and probably expand subsection 1.1.

Materials and Methods

It is provided sufficient information about the participants and the instruments used to measure the variables included in this study. Only some minor questions should be improved:

In the description of the Intolerance of Uncertainty Scale, it would be convenient to include the coefficient of internal consistency (alpha de Cronbach) of the two factor of this scale, and also indicate examples of specific items of each factor.

I think there is a typo of the lines 126, and lines 132-133. I think some words are missing in these sentences.

Results

In Table 1, “sd” should be in capital: SD

Lines 175-183, and Table 2: “p=0.001” should be changed to “p <.001”

Lines 187-193, and lines 199-204: “p=0.00” should be changed to “p <.01”

Discussion

Although some limitations of this study are indicated by authors, it should also be noted that an important limitation is the small sample size. The number of participants is small and this could affect the results, as well as the way in which the sample was accessed (online). These limitations should also be commented.

Reviewer 3 Report

This write up needs to be re-presented. 

It was difficult to read from the abstract to the discussion/conclusion.

Data collection was done via a social media platform - raises concerns regarding privacy and ethics.

Was there an Ethics approval?
